# Room temperature chirality switching and detection in a helimagnetic MnAu$_2$ thin film

Hidetoshi Masuda [1] ✉, Takeshi Seki [1] ✉, Jun-ichiro Ohe [2], Yoichi Nii [1,3], Hiroto Masuda [1], Koki Takanashi[1,4,5] & Yoshinori Onose [1] ✉

Helimagnetic structures, in which the magnetic moments are spirally ordered, host an internal degree of freedom called chirality corresponding to the handedness of the helix. The chirality seems quite robust against disturbances and is therefore promising for next-generation magnetic memory. While the chirality control was recently achieved by the magnetic field sweep with the application of an electric current at low temperature in a conducting helimagnet, problems such as low working temperature and cumbersome control and detection methods have to be solved in practical applications. Here we show chirality switching by electric current pulses at room temperature in a thin-film MnAu$_2$ helimagnetic conductor. Moreover, we have succeeded in detecting the chirality at zero magnetic fields by means of simple transverse resistance measurement utilizing the spin Berry phase in a bilayer device composed of MnAu$_2$ and a spin Hall material Pt. These results may pave the way to helimagnet-based spintronics.

In solids, an internal degree of freedom emerges upon a phase transition involving symmetry breaking. For example, a ferroelectric phase transition breaks the space inversion symmetry and induces spontaneous electric polarization. The two states with positive and negative polarizations are completely degenerate in the absence of electric fields, and therefore the polarization can be viewed as an internal degree of freedom. The application of electric fields can switch the sign of electric polarization. Such a controllable internal degree of freedom is useful for memorizing information and is therefore applicable to memory storage devices. Indeed, random-access memories based on ferroelectrics have been fabricated and are commercially available[1,2].

A more important example is a ferromagnet. In the ferromagnetic state, the time-reversal symmetry is broken, and the magnetization is the internal degree of freedom, which can be controlled by a magnetic field. Hard disk drives utilize ferromagnets, and magnetic random-access memory (MRAM) has also been developed[3,4]. One of the major obstacles for high-density MRAM is stray fields. As the bit scale is decreased, the magnetizations of separated ferromagnets do not work as independent degrees of freedom owing to the entanglement caused by stray fields.

In order to resolve this issue, spintronics based on antiferromagnets is currently attracting considerable attention[5–8]. A helical magnet[9] is one form of antiferromagnet that has unique characteristics: Mirror symmetry is broken, and the chirality works as an internal degree of freedom unless the crystal structure is noncentrosymmetric. The chirality does not couple to the magnetic field and is invariant under any translation and rotation. In order to reverse it, one has to first straighten the spin direction and wind it reversely. In other words, the helimagnetic memory is topologically protected in that way and should be stable even in a very small device. Therefore, this seems to be a desirable degree of freedom for next-generation magnetic storage. Nevertheless, the chirality in conducting helimagnets that are compatible with spintronic devices had been uncontrollable until recently, whereas the chirality in insulating helimagnets is known to be controllable with an electric field[10,11].

Recently Jiang et al.[12] showed that the degeneracy relevant to the chirality is lifted by the simultaneous application of magnetic fields

[1]Institute for Materials Research, Tohoku University, Sendai, Japan. [2]Department of Physics, Toho University, Funabashi, Japan. [3]PRESTO, Japan Science and Technology Agency, Kawaguchi, Japan. [4]Center for Science and Innovation in Spintronics, Tohoku University, Sendai, Japan. [5]Present address: Advanced Science Research Center, Japan Atomic Energy Agency, Ibaraki, Japan. ✉e-mail: hidetoshi.masuda.c8@tohoku.ac.jp; takeshi.seki@tohoku.ac.jp; yoshinori.onose.b4@tohoku.ac.jp

and electric currents with a mechanism totally different from the insulating case. Figure 1a illustrates a conduction electron and localized moments in a helimagnet. The spin of the conduction electrons is aligned to the localized moments owing to the $s-d$ exchange coupling $J_{sd}$. When a conduction electron propagates along the propagation vector of the helimagnet, the spin rotates around the propagation vector, giving rise to the spin transfer torque on localized magnetic moments. The adiabatic spin transfer torque rotates the magnetic structure around the propagation vector and non-adiabatic and damping torques deform it conically[13–15]. The net magnetization direction of the conical magnetic structure depends on the chirality. By additionally applying a magnetic field parallel to the conical magnetization, the chiral degeneracy is lifted. In other words, the favored chirality depends on whether the electric current is parallel or antiparallel to the magnetic field as illustrated in Fig. 1b. Based on this mechanism, Jiang et al. demonstrated chirality control at temperatures around 50 K in a microfabricated single crystal piece of MnP helimagnetic conductor. However, the method used to control the chirality involved a complex sequence of current and magnetic field changes. For practical devices, a more straightforward approach for switching the chirality is needed. It must operate at room temperature, and in thin films to be suitable for industrial fabrication, with easy readout of the chirality.

In this work, we show room-temperature chirality control in a thin-film $MnAu_2$ helimagnet by solely applying an electric current pulse in a magnetic field. Moreover, we show the chirality can be probed by the transverse resistance at zero magnetic fields in a bilayer device composed of $MnAu_2$ and a spin Hall material Pt, which originates from the spin Berry phase (Fig. 1c).

## Results

### Properties of MnAu₂ thin film sample

$MnAu_2$ crystallizes into a centrosymmetric tetragonal crystal structure with the space group $I4/mmm$[16]. The Mn magnetic moments show a helical magnetic order with a helical plane perpendicular to the propagation vector $\mathbf{q} = (0, 0, 0.28)$ in the reciprocal lattice unit, corresponding to the helical pitch of 3.1 nm[17]. The transition temperature is reported to be as high as $T_c = 360$ K[18]. We prepared single-crystal films of $MnAu_2$ with a thickness of 100 nm on hexagonal $ScMgAlO_4$ (10–10) substrates. X-ray diffraction (XRD) measurements revealed that the $MnAu_2$ thin films are stacked along the [110] direction so that the helical propagation vectors are parallel to the thin films (Fig. 2a, b, see "Methods" section and Supplementary Fig. 3 for more detail). The magnetic susceptibility $M/H$ along the [001] direction shows a clear kink at 335 K (Fig. 2c). The resistivity $\rho$ shows metallic temperature dependence with a cusp-like anomaly also around 335 K, as shown in Fig. 2d. The anomalies can be ascribed to the helical transition temperature[18–20], which is slightly lower than the reported value presumably because of epitaxial strain. Similar kinks appear in the magnetic field dependences of magnetization and resistivity. Figure 3a shows the magnetization and magnetoresistance curves for $H \parallel [001]$ at 300 K, in which clear kinks are observed at 1.5 T. Above the kink field, the magnetization is saturated, which indicates that they are caused by the transition from the helical phase to the induced-ferromagnetic (FM) phase[18,20]. The transition field increases as the temperature is lowered, as shown in Fig. 2e.

### Chirality control by magnetic field sweep

We first demonstrate chirality control at room temperature by means of a magnetic field sweep with the application of an electric current, similar to Jiang et al.[12] A magnetic field $H_0 = \pm 3$ T and a dc electric current $j_0$ were first applied along the helical propagation vector. The magnetic field and the electric current were parallel ($H_0 > 0, j_0 > 0$ or $H_0 < 0, j_0 < 0$) or antiparallel ($H_0 > 0, j_0 < 0$ or $H_0 < 0, j_0 > 0$). Then, we swept the magnetic field to 0 T and turned off the dc electric current.

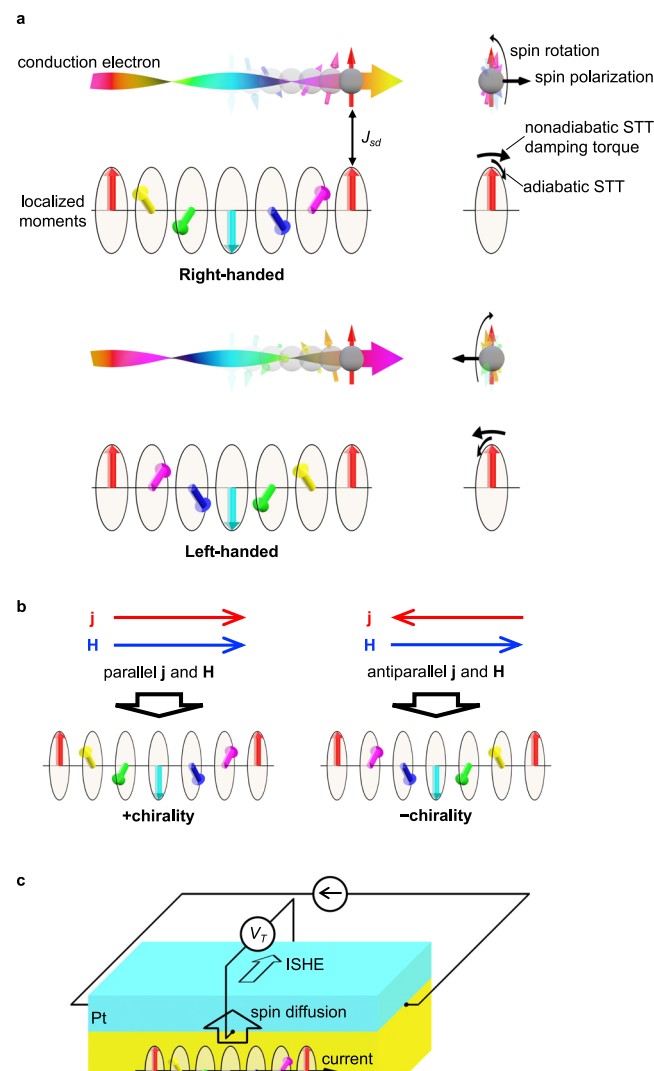

**Fig. 1 | Concept of chirality control and chirality-dependent transverse resistance in helimagnets. a** Schematic illustration of the effects of electron propagation along the propagation vector through a helimagnetic structure. The spin of conduction electron is rotated depending on the chirality, which induces the spin polarization along the wave vector. Reciprocally, the adiabatic spin-transfer torque (STT), non-adiabatic STT and the damping torque are exerted on the localized moments. The adiabatic STT rotates the magnetic structure within the helical plane, while the non-adiabatic STT and damping torques deform it conically. **b** Schematic illustration of the chirality control. The chirality corresponding to the handedness of the magnetic spiral is controlled by the electric current ($j$) and the magnetic field ($H$), depending on whether $j$ and $H$ are parallel or antiparallel. **c** Schematic illustration of the measurement setup for the transverse resistance in the helimagnet MnAu₂/Pt bilayer device. The substrate is not shown for clarity. Electric current is applied parallel to the helimagnetic propagation vector (MnAu₂ [001] direction). The electric current in the helimagnetic MnAu₂ layer induces a chirality-dependent spin polarization. The accumulated spin polarization diffuses into the Pt layer, inducing the transverse voltage $V_T$ by means of the inverse spin Hall effect (ISHE).

After this control procedure, we read out the controlled chirality utilizing the nonreciprocal electronic transport (NET)[21–27], which is a field-asymmetric component of the second-harmonic resistivity $\rho^{2\omega}$ measured by an ac electric current, $\rho^{2\omega}_{asym}(H) = [\rho^{2\omega}(+H) - \rho^{2\omega}(-H)]/2$. Note that $\rho^{2\omega}$ has the same unit as the ordinary resistivity and is proportional to the applied current (see Supplementary Fig. 5). The NET shows up only when the inversion and time-reversal symmetries are simultaneously broken. It reverses its sign upon a space-inversion or time-

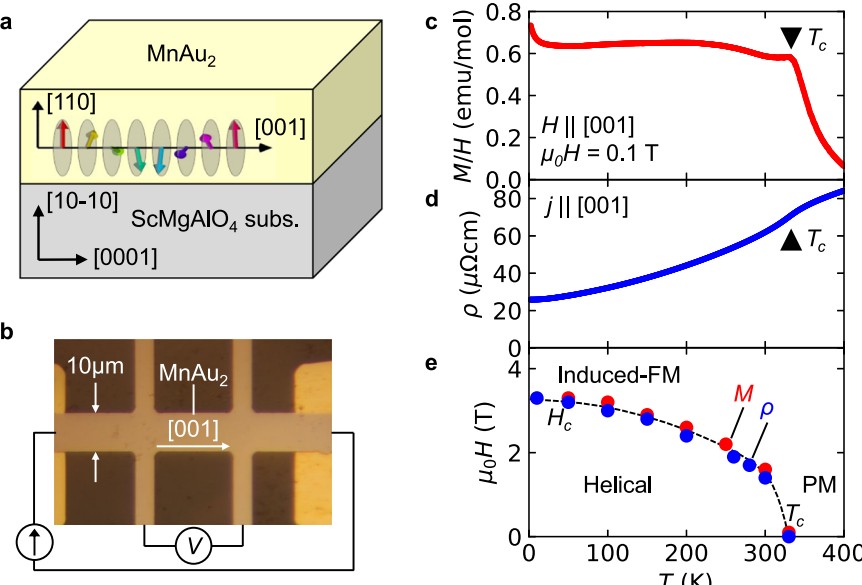

**Fig. 2 | Properties of the MnAu₂ thin film. a** Schematic illustration of the MnAu₂ thin film sample. MnAu₂ was epitaxially grown along the [110] direction with a thickness of 100 nm on a ScMgAlO₄ (10−10) substrate. The propagation vector of the helimagnetic structure is parallel to the [001] direction of MnAu₂ in the sample plane. The Ta cap layer (2 nm) is not shown for clarity. **b** Optical microscope image of the sample device. **c** Temperature $T$ dependence of the magnetic susceptibility $M/H$, which is obtained by the magnetization $M$ divided by the magnetic field $H$. The magnetic field as large as 0.1 T is applied along the [001] direction. **d** $T$ dependence of the resistivity $\rho$. The electric current is applied along the [001] direction. **e** Magnetic phase diagram in the $H$–$T$ plane for the MnAu₂ thin sample. The magnetic transition points are obtained by the magnetization and resistivity measurements. PM and induced FM denote the paramagnetic and field-induced ferromagnetic states, respectively. $H_c$ denotes the transition field from the helical phase to the induced-FM phase. The dashed line is merely a guide for the eyes.

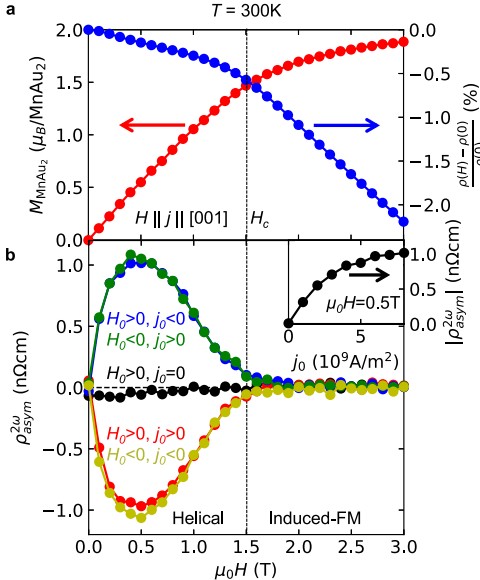

**Fig. 3 | Chirality control by magnetic field sweep with application of electric current. a** Magnetic field $H$ dependence of the magnetization $M_{MnAu2}$ and the magnetoresistance $[\rho(H) − \rho(0)]/\rho(0)$ at $T = 300$ K. The linear diamagnetic contribution from the substrate is subtracted from the magnetization. The vertical dashed line denotes $H_c$. **b** Magnetic field dependence of $\rho^{2\omega}_{asym}(H)$ at 300 K after the chirality control by the magnetic field sweep from the magnetic field $H_0$ ($H_0 = \pm 3$ T) with the application of electric current $j_0 = 0, \pm 8.0 \times 10^9$ A/m². $\rho^{2\omega}_{asym}(H)$ is the anti-symmetric part of the second-harmonic electrical resistivity $\rho^{2\omega}(H)$, viz., $\rho^{2\omega}_{asym}(H) = [\rho^{2\omega}(+H) − \rho^{2\omega}(−H)]/2$. $\rho^{2\omega}(+H)$ and $\rho^{2\omega}(−H)$ were independently measured just after the chirality control. The ac electric current used for the $\rho^{2\omega}(H)$ measurement is $2.0 \times 10^9$ A/m². The inset shows the $j_0$ dependence of $|\rho^{2\omega}_{asym}(H)|$ at 0.5 T.

reversal operation (see Supplementary Information, section 2). Since the chirality is reversed upon the space-inversion operation, the sign of NET probes the chirality[12,21,24–27]. In order to obtain $\rho^{2\omega}_{asym}(H)$, we measured $\rho^{2\omega}(H)$ while increasing $H$ from 0 to 3 T and that also while decreasing $H$ from 0 to −3 T and calculated the difference. Figure 3b shows the magnetic field dependence of the NET signal $\rho^{2\omega}_{asym}(H)$ after the field sweep chirality control with $H_0 = \pm 3$ T and $j_0 = 0, \pm 8.0 \times 10^9$ A/ m². While $\rho^{2\omega}_{asym}(H)$ was almost negligible for $j_0 = 0$, finite $\rho^{2\omega}_{asym}(H)$ was observed for the other data. The magnitude steeply increases as the field magnitude is increased from 0 T. It shows a maximum around 0.5 T and almost vanishes above the ferromagnetic transition field. We confirm the magnetic field angle dependence of $\rho^{2\omega}_{asym}(H)$ is consistent with the chiral symmetry (see Supplementary Figs. 5, 6 for more detail). Importantly, the sign of the NET signal depends on whether $H_0$ and $j_0$ are parallel or antiparallel, confirming that the chirality was controlled successfully. The inset for Fig. 3b shows the $j_0$ dependence of the NET signal. The NET signal monotonically increases with $j_0$ and saturates around $j_0 = 6.0 \times 10^9$ A/m², suggesting that the controlled volume fraction approaches unity.

## Chirality switching by electric current pulses

Then, we performed the chirality control by the application of electrical current pulses. A theory in the literature[28] showed that the chirality can be controlled by the application of an electrical current pulse under a small magnetic field in the helimagnetic state when the magnitude of the electric current is much larger. The simple switching largely increases the availability of the chiral degree of freedom in spintronic devices. To experimentally demonstrate the chirality switching, we first swept the magnetic field from the high field to zero without an electric current so that the two chiral domains are almost equally distributed (see Fig. 3b), and then applied positive and negative electrical pulses with a duration of 1 ms alternately every 15 min at 0.5 T while measuring the second-harmonic resistivity. Figure 4a shows the time dependence of the second-harmonic resistivity change

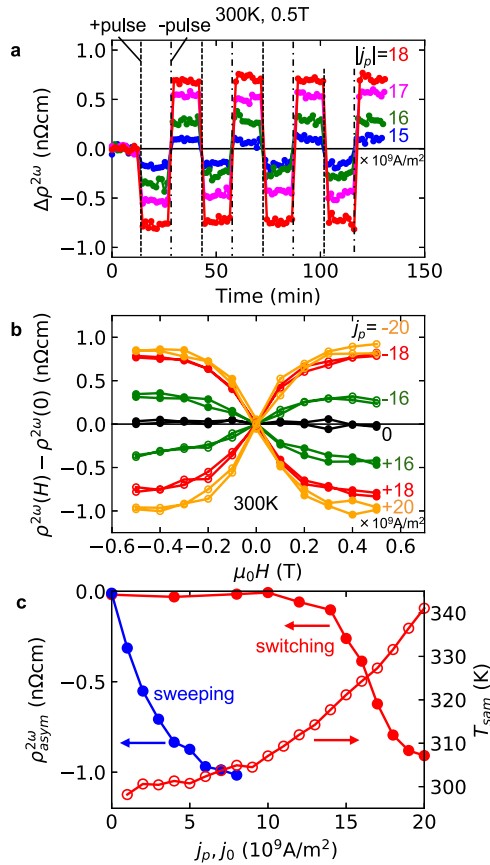

**Fig. 4 | Chirality switching. a** Temporal variation of second-harmonic resistivity change $\Delta\rho^{2\omega}$ at 300 K and 0.5 T with the alternate application of positive and negative electric current pulses with a duration of 1 ms. Before the measurement, the magnetic field is swept from 3 to 0 T without electric current so that two different chirality domains are equally distributed. The magnitudes of the pulse currents are $|j_p|$ = 15 (blue), 16 (green), 17 (magenta), and 18 (red) × $10^9$ A/m². **b** Magnetic field dependence of $\rho^{2\omega}$ at 300 K after application of the current pulses at 0.5 T. The pulse currents are $j_p$ = 0 (black), +16 × $10^9$ A/m² (green, filled), −16 × $10^9$ A/m² (green, open), +18 × $10^9$ A/m² (red, filled), −18 × $10^9$ A/m² (red, open), +20 × $10^9$ A/m² (orange, filled) and −20 × $10^9$ A/m² (orange, open). Before the measurement, the magnetic field is swept from 3 to 0 T without an electric current, similarly to the case of Fig. 4a. **c** $\rho^{2\omega}_{asym}$ at 300 K and 0.5 T as a function of $j_p$ (red-filled circles). The $j_0$ dependence of $\rho^{2\omega}_{asym}$ in the case of field sweep control is reproduced from Fig. 3b for comparison (blue-filled circles). The red-open circles indicate the sample temperature $T_{sam}$ during the applied current pulse $j_p$ estimated from the sample resistance (see Supplementary Fig. 8 for more detail).

$\Delta\rho^{2\omega}$ after the zero current field sweep. When the pulse current $j_p$ was larger than $15 \times 10^9$ A/m², a discontinuous change of $\Delta\rho^{2\omega}$ appeared. The magnitude of the discontinuous change increases as the current is increased, and the negative pulse reversed $\Delta\rho^{2\omega}$. We observed the alternating change of $\Delta\rho^{2\omega}$ several times. Note that the magnitude of $\Delta\rho^{2\omega}$ is not increased by the multiple pulse application, and that the linear resistivity does not show the discontinuous change (see Supplementary Fig. 7). To confirm that such a discontinuous change of the second-harmonic resistivity corresponds to the chiral domain change, we measured the field dependence of the second-harmonic resistivity after the application of an electric current pulse at 0.5 T. For this experiment, we also performed a zero-current magnetic field sweep from 3 to 0 T before the pulse application. After the pulse application, we decreased the magnetic field to −0.5 T and restored it to +0.5 T while measuring the second-harmonic resistivity. The measured second-harmonic resistivity data are shown in Fig. 4b. In this figure, the difference from the zero-field value is plotted just for clarity. The

asymmetric field dependence is clearly observed, and its sign depends on that of the current pulse. These results show that the discontinuous change of the second-harmonic resistivity is certainly caused by the chiral domain change.

We discuss the $j_p$ dependence of $\rho^{2\omega}_{asym}$ estimated from the magnetic field dependence after the pulse application (Fig. 4c). $\rho^{2\omega}_{asym}$ sharply increases around $1.4 \times 10^{10}$ A/m² and saturates around $1.9 \times 10^{10}$ A/m². For comparison, we reproduce the electric current dependence of $\rho^{2\omega}_{asym}$ for the field sweep control case. A much larger electric current is needed for the switching but the magnitude of the controlled $\rho^{2\omega}_{asym}$ is comparable with the sweeping case, suggesting that the controlled volume fraction approaches unity also for the pulse case. Thus, chiral domain switching is achieved for this thin film sample. Nevertheless, it should be noted that the heating effect seemed to assist the switching phenomenon. The sample resistance estimated from the voltage during the pulse application is larger than that at 300 K, indicating heating of the sample (see Supplementary Fig. 8). The estimated sample temperature $T_{sam}$ is plotted against $j_p$. The threshold current density $j_p = 1.4 \times 10^{10}$ A/m² corresponds to the sample temperature of 318 K, which is lower than the transition temperature 335 K. It gradually increases and exceeds the transition temperature 335 K around $2 \times 10^{10}$ A/m². In this case, the chirality was erased by the increase of temperature but perhaps controlled again at a moment when the magnitude of the electric current was decreased. When the nominal experimental temperature is decreased to 290 K the $j_p$ dependence is also shifted, which suggests that the heating effect certainly contributes to the chiral switching phenomenon (see Supplementary Fig. 9). The critical current of chirality domain control is expected to decrease with increasing temperature toward $T_c = 335$ K. Presumably, it becomes low enough around 318 K for the chirality switching, and finite $\rho^{2\omega}_{asym}$ is observed above $1.4 \times 10^{10}$ A/m². Traversing the helical transition temperature is not mandatory in this phenomenon.

### Chirality-dependent transverse resistance in a MnAu₂/Pt bilayer device

Finally, we show that the chirality can be probed by a simple transverse resistance measurement in a MnAu₂/Pt bilayer device (Fig. 1c). The origin of the transverse resistance can be explained by the following two steps. First, an electric current along the helimagnetic wave vector induces spin accumulation depending on the chirality in the helimagnetic MnAu₂ layer. Next, the accumulated spin is diffused to the Pt layer and induces a transverse voltage by means of the inverse spin Hall effect (ISHE)[29–31]. While the mechanism of the second step is well known[32,33], we explain the first step microscopically below. As shown in Fig. 1a, when a conduction electron propagates along the propagation vector of the helimagnet, the spin rotates around the propagation vector with a sense that depends on the chirality. The chirality-dependent spin rotation affects the dynamics of the conduction electron through the fictitious vector potential (Berry connection) $\mathbf{A} = (i\gamma q\hat{\sigma}_x, 0, 0)$, where the $x$-axis is along the propagation vector of the helimagnet, $q$ is the magnitude of the propagation vector, $\gamma = \pm 1$ is the chirality handedness, and $\hat{\sigma}_x$ is the Pauli matrix[34,35]. Therefore, when an electric current $j = (j_x, 0, 0)$ is applied, the current-dependent energy

$$\Delta E \sim <\mathbf{p}\cdot\mathbf{A}> \sim \mathbf{j}\cdot\mathbf{A}\gamma q\hat{\sigma}_x j_x \qquad (1)$$

gives rise to spin polarization depending on the chirality. This phenomenology was numerically investigated for a helimagnet induced by the Dzyaloshinskii–Moriya interaction with the fixed chirality in the literature[34]. We confirmed the validity also for a variable-chirality helimagnetic model (see Supplementary Information, Section 1).

To experimentally demonstrate the transverse resistance originating from the chirality-dependent spin accumulation, we prepared bilayer devices consisting of MnAu₂ and Pt on hexagonal ScMgAlO₄

substrates (see "Methods" and Supplementary Figs. 4, 10). Similarly to the MnAu$_2$ sample without Pt shown above, the helical propagation vector along the [001] direction is parallel to the film plane. The Pt layer with a thickness of 10 nm was deposited on MnAu$_2$. When an electric current $I$ is applied along the MnAu$_2$ [001] direction, the chirality-dependent accumulated spin moment should diffuse to the Pt layer, and a transverse voltage $V_T$ is expected to emerge owing to the ISHE as schematically shown in Fig. 1c. Therefore, the chirality should be probed by the transverse resistance $R_T = V_T/I$.

Figure 5 demonstrates the observation of chirality-dependent transverse resistance at 260 K in the MnAu$_2$/Pt bilayer device. We first reproduced the field sweep chirality control and the NET signal for the bilayer device for comparison with the transverse resistance. Figure 5a, b shows the magnetic field $H$ dependence of $R^{2\omega}_{asym}$ at 260 K measured after the magnetic field sweep from $H_0 = \pm 5$ to 0 T with the application of a dc electric current $I_0 = \pm 8$ mA ($H_0 > 0$ for Fig. 5a and $H_0 < 0$ for Fig. 5b). Finite NET signals were clearly observed in the helimagnetic state $|H| < H_c = 1.9$ T (see Supplementary Fig. 10), and their signs depended on whether the control field $H_0$ and control current $I_0$ are parallel or antiparallel, confirming that the chirality control was successfully reproduced also for this sample. In Fig. 5g we plot the averaged NET of $\Delta R^{2\omega}_{asym} (|I_0|) = (R^{2\omega}_{asym}(+I_0) - R^{2\omega}_{asym}(-I_0))/2$ as a function of $|I_0|$. The magnitude of $\Delta R^{2\omega}_{asym}$ monotonically increases with increasing $|I_0|$ and tends to saturate. Figure 5c, d shows the $H$ dependence of the transverse resistance $R_T$ after the chirality control procedure with the magnetic field $H_0$ and the electric current $I_0$

at 260 K. Note that the data for $H > 0$ T and $H < 0$ T were separately measured by sweeping the magnetic field from 0 T just after the chirality control procedure. The main contribution to $R_T$ seems to be from trivial effects such as longitudinal resistance, the Hall effect, and the planar Hall effect arising from the misalignments of voltage electrodes and the magnetic field. The slight difference between the positive and negative $H_0$ (Fig. 5c, d, respectively) for the same chirality should be ascribed to the magnetic hysteresis. Nevertheless, the apparent difference between the positive and negative $I_0$ for the same $H_0$ cannot be ascribed to any trivial effects. Because it is reversed by the reversal of $H_0$, the nontrivial component seems to depend on whether $H_0$ and $I_0$ are parallel or antiparallel, as indicated by the dotted lines in Fig. 5c, d. At zero field, $R_T$ is around −1.48 mΩ for one chiral state ($H_0 > 0$, $I_0 > 0$ or $H_0 < 0$, $I_0 < 0$) and −1.57 mΩ for the other ($H_0 > 0$, $I_0 < 0$ or $H_0 < 0$, $I_0 > 0$). Therefore, this quantity seems to be useful for probing the chirality at zero magnetic field. To examine the chirality dependence of $R_T$, we calculate the difference between $R_T$ for positive and negative $I_0$ divided by 2, that is, $\Delta R_T = (R_T(+I_0) - R_T(-I_0))/2$ (Fig. 5e ($H_0 > 0$) and Fig. 5f ($H_0 < 0$)). $\Delta R_T$ shows a maximum around 0 T and decreases with increasing $H$ or decreasing $H$ from $H = 0$ T. Then, it almost vanishes in the induced-ferromagnetic phase. The sign is reversed by the reversal of $H_0$. These properties indicate that $\Delta R_T$ reflects helimagnetic chirality. To confirm this, we have investigated the $|I_0|$ dependence of $\Delta R_T$ (Fig. 5h). It is quite similar to the $|I_0|$ dependence of $\Delta R^{2\omega}_{asym}$ (Fig. 5g). In fact, $\Delta R_T$ is proportional to $\Delta R^{2\omega}_{asym}$, as shown in Fig. 5i. That is to say, $\Delta R_T$ certainly reflects the

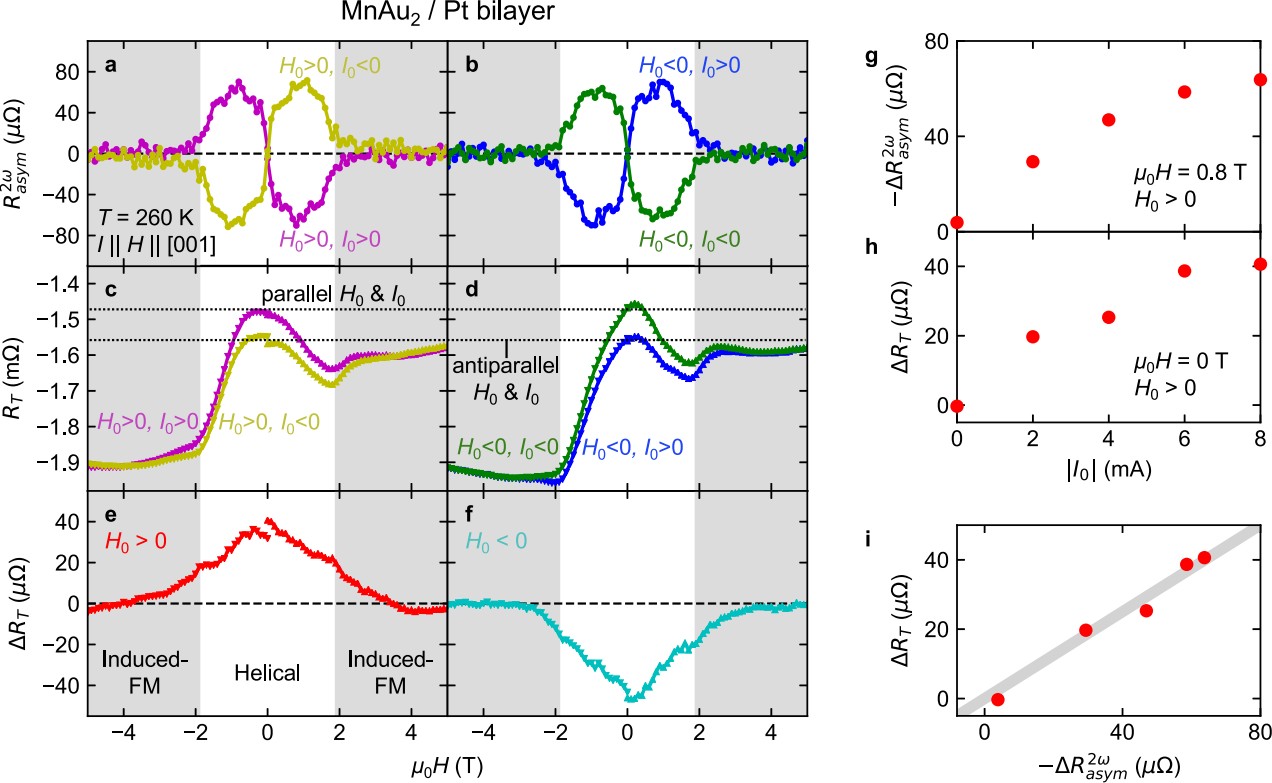

**Fig. 5 | Observation of the chirality-dependent transverse resistance in the MnAu$_2$/Pt bilayer sample. a, b** Magnetic field $H$ dependence of $R^{2\omega}_{asym}$ at 260 K after the field-sweep chirality control with the magnetic field $H_0 = \pm 5$ T and electric current $I_0 = \pm 8.0$ mA for the MnAu$_2$/Pt bilayer sample. The results for $H_0 > 0$ and $H_0 < 0$ are shown in (**a, b**), respectively. While the data of $H < 0$ are merely copies of the $H > 0$ data, we plot the $H < 0$ data just for clarity. The gray shading represents the induced-ferromagnetic (FM) phase. The ac electric current used for the $R^{2\omega}(H)$ measurement is 2.0 mA. **c, d** Magnetic field dependence of the transverse resistance $R_T$ after the chirality control with $H_0$ and $I_0$ for the MnAu$_2$/Pt bilayer sample.

The results for $H_0 > 0$ and $H_0 < 0$ are shown in (**c, d**), respectively. The horizontal dotted lines indicate the maxima of $R_T$ for the parallel and antiparallel $H_0$ and $I_0$. **e, f** Magnetic field dependence of the chirality-dependent component of the transverse resistance estimated from the relation $\Delta R_T = (R_T(+I_0) - R_T(-I_0))/2$ for $H_0 > 0$ and $H_0 < 0$ are shown in (**e, f**), respectively. **g** $|I_0|$ dependence of $-\Delta R^{2\omega}_{asym}$ at 0.8 T for the MnAu$_2$/Pt bilayer sample. Here, $\Delta R^{2\omega}_{asym}$ is the averaged NET signal $(R^{2\omega}_{asym}(+I_0) - R^{2\omega}_{asym}(-I_0))/2$. **h** $|I_0|$ dependence of $\Delta R_T$ at 0 T for the MnAu$_2$/Pt bilayer sample. **i** $\Delta R_T$ at 0 T is plotted against $-\Delta R^{2\omega}_{asym}$ at 0.8 T. The gray line indicates the linear relation.

helimagnetic chirality, which should be caused by the aforementioned spin accumulation mechanism. To further examine the mechanism, we also performed similar measurements on the $MnAu_2$/Cu/Pt trilayer film sample and the $MnAu_2$/W bilayer film sample (Supplementary Figs. 13, 14). In the case of the $MnAu_2$/Cu/Pt trilayer film sample, $\Delta R_T$ is rather enhanced, which indicates that the origin of $\Delta R_T$ cannot be ascribed to some artifact originating from the $MnAu_2$/Pt interface. For the $MnAu_2$/W bilayer film sample, the sign of $\Delta R_T$ is opposite to that of the $MnAu_2$/Pt bilayer device, being consistent with the opposite signs of spin Hall conductance in Pt and W[32]. While we analyzed $\Delta R_T$ in detail at 260 K since it was almost maximized, $\Delta R_T$ is observed in a wide temperature range including room temperature (Supplementary Fig. 11). Therefore, $\Delta R_T$ is useful as a chirality probe even at room temperature and does not require any magnetic field.

## Discussion

In summary, we have demonstrated the chirality switching and detection in a room-temperature helimagnet $MnAu_2$. The chirality degree of freedom is quite robust against magnetic disturbances and is free from the stray field problem. Similar electric current controls of the magnetic structure were performed for some antiferromagnets[36–46]. Some of them utilized symmetry breaking[36–38] and the others the spin current injection from the interface with spin Hall materials[39–46]. The threshold current density of $j_p = 1.4 \times 10^{10}$ A/m$^2$ for the chirality switching in the present work is relatively low compared to these works. Further reducing the electric current and extending the working temperature of chirality switching may be achieved by weakening the magnetic anisotropy by means of a suitable choice of substrate (see Supplementary Information, section 3). The transition temperature of $T_c = 335$ K might be low for practical applications. In a bulk polycrystal, the helimagnetic transition temperature is as high as 360 K[17,18]. The lower transition temperature of 335 K in our thin film is possibly due to the epitaxial strain. The increase of the transition temperature up to 360 K or even higher should be feasible by improving the thin film growth condition. The transition temperature may be increased further by some chemical adjustments. For example, for another helimagnet CrAs, its transition temperature can be increased by 90 K (from 250 to 340 K) by the 14% substitution of As by Sb[47]. Therefore, the increase of transition temperature up to around 450 K may not be too ambitious. In addition, the thin film form of the sample enables us to utilize the interface-based functionality, the chirality-dependent transverse resistance. This result demonstrates current-induced spin accumulation originating from the Berry phase effect in the coplanar helimagnetic state, which is distinct from the well-known Berry phase effect of the emergent magnetic field in noncoplanar magnetic textures such as a skyrmion lattice[48–50]. A similar spin-polarization phenomenon known as chirality-induced spin selectivity (CISS) has been observed in chiral molecules and crystals[51,52]. The present observation can be viewed as a full magnetic analog of CISS. In a helimagnet with a nonchiral crystal structure, the magnetic CISS enables us to read out the chiral information even in the absence of magnetic fields. Because the transverse voltage is induced at the interface, the magnitude of transverse voltage is expected to become larger as the thickness of the thin film is reduced. If a high-quality ultrathin film can be fabricated, the magnitude may satisfy the requirements for practical application as a probe of chirality-based magnetic memory. Thus, the present result is certainly a milestone for new technology helimagnetic spintronics.

## Methods

### Sample fabrication

Epitaxial films of $MnAu_2$ with a thickness of 100 nm were deposited on $ScMgAlO_4$ (10–10) substrates by magnetron sputtering from Mn and Au targets at 400 °C. To fabricate the sample for the measurements shown in Figs. 3, 4, one of the $MnAu_2$ films was in situ covered by a Ta cap layer with a thickness of 2 nm and annealed at 600 °C for 1 h. The thin Ta layer may be oxidized and less conductive, but we expected the $MnAu_2$ layer was almost free from oxidization, and the sample can be regarded as a single-layer $MnAu_2$ sample. To fabricate the $MnAu_2$/Pt bilayer device, a Pt layer with the thickness of 10 nm was deposited on a $MnAu_2$ thin film at room temperature after annealed at 600 °C for 1 h. The XRD results indicate the epitaxial growth of $MnAu_2$ (110) on the $ScMgAlO_4$ (10–10) substrate, where the $MnAu_2$ [001] and the $ScMgAlO_4$ [0001] directions are parallel to each other (Supplementary Figs. 3 and 4). While a previous paper reports the growth of polycrystalline and multi-domain $MnAu_2$ films[53], the suitable choice of substrate enabled the epitaxial growth of single crystalline films. The Pt layer in the bilayer device was revealed to be parallel to the (111) plane. The Pt layer is robust against oxidization and thicker than the Ta cap case. Therefore, the Pt/$MnAu_2$ sample can work as a bilayer device.

### Magnetization measurement

The magnetization measurements were performed using a Magnetic Property Measurement System (Quantum Design).

### First- and second-harmonic resistance measurement

For the resistance measurements, the thin film sample was patterned into Hall bar devices by photolithography and Ar plasma etching. Optical micrograph of the Hall bar device is shown in Fig. 2b. The direction of the electric current was parallel to the $MnAu_2$ [001] direction. The width of the Hall bars was 10 μm, and the distance between two voltage electrodes for the resistivity measurement was 25 μm. Electrical contacts were made by photolithography and electron beam evaporation of Ti (10 nm)/Au (100 nm). The longitudinal, transverse, and second-harmonic longitudinal ac resistances were measured by the lock-in technique with an ac electric current frequency of 11.15 Hz and an amplitude of 2 mA in a superconducting magnet or a Physical Property Measurement System (Quantum Design). For the $MnAu_2$/Pt bilayer device shown in Fig. 5, we simultaneously measured the transverse resistance and second-harmonic longitudinal resistance.

## Data availability

The data that support the findings of this study are available from the corresponding authors upon reasonable request. Source data are provided with this paper.

## Code availability

The codes that support the findings of this study are available from the corresponding authors upon reasonable request.

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

## Acknowledgements

The authors thank T. Sasaki and T. Endo for their help in doing the film deposition. The film deposition and device fabrication were carried out at the Cooperative Research and Development Center for Advanced Materials, IMR, Tohoku University. This work was partially supported by JSPS KAKENHI grant numbers JP20H00299 (T.S.), JP20K03828 (Y.N.), JP21H01036 (Y.O.), JP21H01799 (J.O.), JP22H04461 (Y.O.), JP23K13654 (Hidetoshi M.), JST PRESTO grant number JPMJPR19L6 (Y.N.), the Mitsubishi Foundation (Y.O.), and the Yazaki memorial foundation for science and technology (Y.O.).

## Author contributions

T.S. grew the film and performed X-ray diffraction under the support of K.T. Hidetoshi M. performed the device fabrication with the use of photolithography and Ar plasma etching, resistance measurements, and magnetization measurements with supports from T.S., Y.N., Hiroto M. and Y.O. J.O. performed the numerical calculation on the electric current-induced spin current. Y.O. conceived the project. Hidetoshi M. and Y.O. wrote the draft with input from T.S., J.O., Y.N., Hiroto M., and K.T.

## Competing interests

The authors declare no competing interests.
