## [Peer Review File · Nature Communications]

Reviewers' Comments:

Reviewer #1:

Remarks to the Author:

The manuscript builds on the recent study by some of the same authors, Ref. 12, which demonstrated that the chirality of spin rotation in a helimagnet could be controlled at cryogenic temperatures by applying magnetic fields and electric currents, and can be measured using 2nd harmonic resistivity. The noteworthy results of the present manuscript are as follows:

- i) By using MnAu₂, a high-temperature helimagnet, the results from Ref. 12 are reproduced at ambient temperatures and above.
- ii) Reproducible chirality switching is demonstrated using millisecond current pulses rather than static currents, which reduces the energy dissipated.
- iii) Detection of the chirality in zero magnetic fields is demonstrated using the inverse spin Hall effect in a Pt overlayer.

Together, these breakthroughs are of considerable significance for spintronics and related fields, as they offer a route to the practical implementation of memory devices based on spin chirality. Moreover, the results are original and provide new insight into the coupling between spin polarization and spin chirality. The experimental results presented are convincing. I strongly support the publication of this work in Nature Communications, subject to consideration of the minor points described below.

1. The amplitude of the ac current used for the 2nd harmonic resistivity measurements should be stated, preferably in the Methods sections. Also it would be useful for the reader if a clearer description could be given for how and why the NET effect arises in these films. For example, a short section could be added to the supplementary materials.

2. It's interesting that chirality switching persists even when the estimated temperature during the pulse is above the transition temperature of the film. What happens when still higher amplitude current pulses are applied?

Reviewer #2:

Remarks to the Author:

The work of Masuda et al. is focused on control and detection of spin chirality in a helimagnet MnAu₂. The authors employ magneto-transport measurement to detect the chiral state by electrical magneto-chiral effect and for the switching of the chirality they use heat assisted "poling scheme". In the last part the author prepare a bilayer of MnAu₂/Pt and they claim to detect the chirality of MnAu₂ thanks to the spin current injected into Pt and converted to transversal voltage via spin Hall effect.

Although I find the chirality switching in helimagnets in principle interesting, the presented manuscript did not convince me about its novelty (details are following) and I lack more careful analysis of the measured data (details are following). Mainly for these reasons I do not recommend for publication.

Major comments:

1) Novelty. The manuscript refers to the previous work of Jiang et al. (ref 12) and the authors claim that Jiang's work "cannot be directly applied to a practical device because of the low working temperature and complex sequence of detection". However, the presented manuscript employed basically the same complex sequence and the critical temperature of 335K is also not suitable for application. In the same time the authors rely on the accessible critical temperature to perform the "heat assisted" chirality reversal. Therefore any discussion about how to increase the critical temperature (Discussion part) should also explain how to solve the reversal. The authors also report novel chirality control by "solely by electric current pulse in a magnetic field". But the authors show that only in the vicinity of the transition temperature- this is exactly the same poling procedure as shown in ref.12 and well known in ferroelectrics. The novel thing would be to use electrical pulse to reverse the chiral domain well below the critical temperature far from the phase

transition, but this was not shown in this manuscript. The last claim of the manuscript – detection via spin current in Pt – would require more careful analysis to be fully convincing.

2) Careful analysis. The authors use a rather complex detection scheme (which should be better described – details in minor comments) which might be vulnerable to spurious signals in case of MnAu₂/Pt bilayer. I consider that especially important since the signals are reaching only nOhm.cm! How did author exclude for example following:

- interfacial polarization of Pt? did they try to insert for example Cu buffer? Or use another heavy metal?
- how much current flows into Pt? Can the Pt inject spin current into MnAu₂? Did the authors repeat the experiment (Fig. 5c,d) above 335K – no difference between the positive and negative I_o is visible?
- how does the R_T signal look without Pt layer?
- what is the anisotropy of the diffused spin signal into Pt?

3) What is exactly the anisotropy of the MnAu₂? The authors claim that “difference in Fig.5c vs d should be ascribed to the magnetic hysteresis. ” But SQUID measurements along the same field [001] in Fig. 3e do not show any hysteresis? Or the SQUID data are somehow offsetted?

4) line 265 – it was not clear to me how Delta R_T “useful chirality probe which does not require magnetic field”, Delta R_T is defined as difference of +I_o and –I_o when magnetic field is applied, correct? Or the magnetic field can be avoided and the traces for example in Fig.5c will look identical only by applying the +I_o and –I_o?

Minor

- 1) Please, add more details about the measurement procedure – for example, line 224 – the electrical current was DC? What was the amplitude (also 2e9 A/m²)? And how does the R_{1w} look? Why was not measured simultaneously panel Fig.5a and Fig.5c?
- 2) line 245: why would Hall and planar Hall arise from misalignment of the voltage electrodes?
- 3) wording could be improved, example: line 223-224 ‘Pt layer deposited on MnAu₂ to a thickness 10nm’
- 4) Fig.5h – not sure if one can call it saturating – maybe measure more points above 8mA or between 6 and 8mA)?
- 5) SI Fig. 10 – DeltaR_T and Delta²w_{asym} – why are they changing sign at 300K?
- 6) SI Fig. 6 – the y scale should be zoomed around the data to clearly show the variation
- 7) Is it necessary to advertise the results heavily in context of application (line 298)? It is very interesting fundamental research that has a long way to application and “chirality based magnetic memory” and many issues to solve (small signals, competition temperature vs heat assisted poling procedure etc).

--- Authors' response to Reviewer 1 ---

We would appreciate Reviewer 1 for supporting the publication in Nature Communications. What follows are our responses to each comment raised by Reviewer 1 and the corresponding revisions in our manuscript.

Reviewer's Comment(1): The amplitude of the ac current used for the 2nd harmonic resistivity measurements should be stated, preferably in the Methods sections. Also it would be useful for the reader if a clearer description could be given for how and why the NET effect arises in these films. For example, a short section could be added to the supplementary materials.

Response(1): Following the comment, we have added the magnitude of ac current for the 2nd harmonic resistivity measurements to the Methods section, and a short section regarding NET to the supplementary information(section 2).

Reviewer's Comment(2): It's interesting that chirality switching persists even when the estimated temperature during the pulse is above the transition temperature of the film. What happens when still higher amplitude current pulses are applied?

Response(2): We expect that the higher amplitude pulse can also control the chirality because it was shown that traversing the transition temperature was not a problem. Nevertheless, we would like not to perform the experiment because the higher current pulse might damage the sample.

--- Authors' response to Reviewer 2 ---

We would thank Reviewer 2 for the comments. What follows are our responses to each comment raised by Reviewer 2 and the corresponding revisions in our manuscript.

Please stop the display of track change when you refer to the line number.

Reviewer's Comment(1): Novelty. The manuscript refers to the previous work of Jiang et al. (ref 12) and the authors claim that Jiang's work "cannot be directly applied to a practical device because of the low working temperature and complex sequence of detection". However, the presented manuscript employed basically the same complex sequence and the critical temperature of 335K is also not suitable for application. In the same time the authors rely on the accessible critical temperature to perform the "heat assisted" chirality reversal. Therefore any discussion about how to increase the critical temperature (Discussion part) should also explain how to solve the reversal. The authors also report novel chirality control by "solely by electric current pulse in a magnetic field". But the authors show that only in the vicinity of the transition temperature- this is exactly the same poling procedure as shown in ref.12 and well known in ferroelectrics. The novel thing would be to use electrical pulse to reverse the chiral domain well below the critical temperature far from the phase transition, but this was not shown in this manuscript. The last claim of the manuscript – detection via spin current in Pt – would require more careful analysis to be fully convincing.

Response(1): Opposed to the reviewer's opinion, we believe that a)high transition temperature, b)chirality switching, and c)zero-field chirality detection in this paper are novel enough for publication in Nature Communications.

a) High transition temperature

While we admit that 335K is not enough for the practical application, exceeding room temperature is an important milestone. That is why many papers titled "Room temperature..." appear in Nature and its sister journals. In this sense, we believe that the reviewer's opinion is too strict.

b) Chirality switching

In ref. 12, the chirality was controlled by sweeping a magnetic field traversing the transition field with the application of electric current. On the other hand, in the present work, we succeeded in controlling the chirality by the application of an electric pulse at a certain magnetic field in a helimagnetic state at room temperature. There are qualitative

differences between the previous and present works; a sweeping magnetic field was used and traversing the transition field was needed in the previous work, but a steady magnetic field was used and traversing the phase transition point was not needed in the present work. We believe that the reviewer's statement "exactly the same poling procedure" is quite misleading.

We honestly admitted the finite effect of heating and carefully examined the impact by measuring the temperature at every moment in the course of the chirality switching. Our conclusion is that there is some heating effect but traversing the transition temperature is not mandatory for the phenomenon of chirality switching as described at the 181st-196th line. The key point of poling procedure employed in ferroelectrics is traversing the phase transition point with the application of an external field. Therefore, there is apparently a certain difference.

As the reviewer suggested, the extension of the working temperature is an important issue. According to our theoretical calculation newly shown in the supplementary information (the 3rd section), the weakening of magnetic anisotropy decreases the critical current of chirality switching. In this sense, a suitable choice of substrate may decrease the magnetic anisotropy and extend the working temperature range of chirality switching. To address this point, we have added a sentence at the 279th-281st in the main text following the reviewer's comment.

c) zero-field chirality detection

This method for probing chirality is based on a theoretical paper (ref. 34). Nevertheless, to the best of our knowledge, any similar experimental paper has not ever been published. The Berry phase mechanism is novel in the light of basic science. The newly obtained functionality of zero-field chirality detection is novel in the light of applied science. We carefully analyzed the experimental data as detailed in the following responses, and are quite sure that the chirality can be probed by the zero-field transverse resistance.

Reviewer's Comment(2): Careful analysis. The authors use a rather complex detection scheme (which should be better described – details in minor comments) which might be vulnerable to spurious signals in case of MnAu2/Pt bilayer. I consider that especially important since the signals are reaching only nOhm.cm! How did author exclude for example following:

- interfacial polarization of Pt? did they try to insert for example Cu buffer? Or use another

heavy metal?

- how much current flows into Pt? Can the Pt inject spin current into MnAu₂? Did the authors repeat the experiment (Fig. 5c,d) above 335K – no difference between the positive and negative I_0 is visible?

- how does the R_T signal look without Pt layer?

- what is the anisotropy of the diffused spin signal into Pt?

Comment(3): What is exactly the anisotropy of the MnAu₂? The authors claim that “difference in Fig.5c vs d should be ascribed to the magnetic hysteresis. “ But SQUID measurements along the same field [001] in Fig. 3e do not show any hysteresis? Or the SQUID data are somehow offseted?

Comment(4): line 265 – it was not clear to me how ΔR_T “useful chirality probe which does not require magnetic field”, ΔR_T is defined as difference of $+I_0$ and $-I_0$ when magnetic field is applied, correct? Or the magnetic field can be avoided and the traces for example in Fig.5c will look identical only by applying the $+I_0$ and $-I_0$?

Response(2),(3)&(4):

These comments are regarding the chirality dependent transverse resistance. To comprehensively respond to these comments, we first respond to the comment (4) and then to the comments (3) & (2).

(Comment (4))

In Figs. 5c&5d, we did so-called control experiments. We first controlled the chirality with an electric current I_0 & a magnetic field H_0 . Then we did the conventional measurement of transverse resistance. I_0 was used only for the chirality control and was quenched before the measurement of transverse resistance. So, it is unrelated to the measurement of transverse resistance. You can find that the transverse resistance is around -1.48 m Ohm for one chiral state ($H_0>0&I_0>0$ or $H_0<0&I_0<0$) and -1.57 m Ohm for the other ($H_0>0&I_0<0$ or $H_0<0&I_0>0$) at zero-field, irrespective of magnetic hysteresis (see horizontal dotted lines). Therefore, the simple transverse resistance at zero field correlates with the chirality. In other words, we can detect the chirality simply by measuring the transverse resistance at zero field. This is the reason why we mentioned “useful chirality probe”. To emphasize this, we have added the description at 253rd-256th line.

The quantity ΔR_T is defined as the difference between R_T after the chirality control with

+ I_0 and that with $-I_0$. I_0 is not the measurement electric current of transverse resistance but the polling electric current for the chirality control, which was quenched before the transverse resistance measurement. The reviewer seemed to misunderstand this point. ΔR_T reflects the chirality difference and is used to examine the correlation between the transverse resistance and chirality. The magnetic field dependence of ΔR_T confirms it was absent in the ferromagnetic state and excludes the magnetic hysteresis as the origin of ΔR_T . In addition, ΔR_T is nicely scaled with the established chirality probe of nonreciprocal electric resistance NET as shown in Fig. 5i. Based on these experimental data, we are quite confident that the transverse resistance can probe the chirality for the Pt/MnAu2 device.

(Response to comment (3))

We did not observe the hysteresis in the magnetization curve and longitudinal resistance as shown below (Fig. R1). Nevertheless, the transverse resistance seems to show hysteresis. This might be because transverse resistance is generally sensitive to magnetic anisotropy as well as uniform magnetization. In fact, the effect of magnetic hysteresis was small in the transverse resistance. The magnetic hysteresis is not what we would like to prove but is just an obstacle. Our analysis carefully treated such a small effect and deduce the relation between the chirality and transverse resistance in a watertight way.

Figure R1 Magnetic field dependences of magnetization (a) and longitudinal resistance

(b) for the MnAu2/Pt bilayer device.

(Response to comment (2))

It is true that the chirality-dependent transverse resistance is not large. We already measured and analyzed it very carefully. As a result, it clearly shows the correspondence between ΔR_T and $R^{2\omega}_{asym}$ in Fig. 5 as mentioned above. We are quite confident that our experimental data certainly shows it. In addition, the comments raised by the reviewer are not directly related to the reliability of the presented experimental data and analysis.

The reviewer calculated the resistivity multiplying dimension factors, but such calculation is irrelevant because this phenomenon is induced by the interface. The magnitude of the signal should be compared in the unit of resistance if one worries about the experimental vulnerability.

>interfacial polarization of Pt? did they try to insert for example Cu buffer? Or use another heavy metal?

ΔR_T is the difference between R_T after the chirality control with $+I_0$ and that with $-I_0$. The conditions for $+I_0$ and $-I_0$ experiments other than the polling electric current I_0 were exactly the same. We are quite sure that any ferromagnetic polarization cannot produce ΔR_T . In this sense, the additional experiment with Cu buffer or another heavy metal, which would take several months, is not needed.

>how much current flows into Pt?

We estimated 20 % of electric current flows into the Pt layer based on the difference between the resistance of the MnAu2/Pt bilayer sample and that of the MnAu2/Ta 2nm cap layer sample (see Supplement Fig. 10).

>Can the Pt inject spin current into MnAu2?

> what is the anisotropy of the diffused spin signal into Pt?

Considering the symmetry, the spin transport with the polarization parallel or antiparallel to the propagation vector of helimagnetic structure is depending on the chirality, but that with the perpendicular spin polarization is not. The spin Hall effect of the Pt layer should inject the spin current with the spin polarization perpendicular to the propagation vector into the MnAu2 layer. The longitudinal resistance may be affected by the transverse spin

current, but it should be insensitive to the chirality.

If the spin current from MnAu₂ to Pt has a polarization perpendicular to the propagation vector, it also affects the longitudinal resistance. The spin Hall effect of MnAu₂ may induce such spin current but it should be, again, independent of the chirality.

We measured the longitudinal resistance simultaneously with the transverse resistance measurement. The result is shown in Fig. R2. We analyzed the data similarly to the transverse resistance, but the chirality-dependent contribution was not observed within the measurement accuracy.

Figure R2 a, b Longitudinal resistance R of the MnAu₂ / Pt bilayer device as a function of magnetic field H at 260 K. These were simultaneously measured with the data in Fig. 5a, c (Fig. 5b, d) after the field-sweep chirality control with the field $H_0 = \pm 5$ T and the current $I_0 = \pm 8.0$ mA.

c, d H dependence of the chirality-dependent component of the resistance estimated from the relation $\Delta R = (R(+I_0) - R(-I_0))/2$ for $H_0 > 0$ and $H_0 < 0$ (c and d, respectively).

>Did the authors repeat the experiment (Fig. 5c,d) above 335K – no difference between

the positive and negative I_0 is visible?

We did the experiment at 340 K, and the difference is negligible. The result is shown in Supplementary Figs. 11f, m.

>how does the R_T signal look without Pt layer?

Our MnAu₂ thin films are always capped by another metal, (in most cases, Ta) to avoid oxidization. The films used in the experiments for Figs. 2, 3, and 4 are also capped by Ta 2nm layer as described in the Methods section. The thin Ta layer may be oxidized and less conductive, but we expected the MnAu₂ layer was almost free from oxidization, and the sample can be regarded as single-layer MnAu₂ sample. In order to respond to this comment, we performed an additional experiment for a Ta 2nm/MnAu₂ sample, similarly to that for Fig. 5. The result is shown below (Fig. R3). While the transverse resistance owing to some misalignment of electrodes is large for this sample, the chirality dependence (ΔR_T) was quite small compared with the MnAu₂/Pt bilayer device. This is another evidence indicating that ΔR_T in the MnAu₂/Pt bilayer device is caused by the spin current from MnAu₂ to Pt layer.

Figure R3 a,b Longitudinal resistance R , **c,d** nonreciprocal 2nd harmonic resistance $R_{\text{asym}}^{2\omega}$, **e,f** transverse resistance, and **g,h** $\Delta R_T = R_T(+I_0) - R_T(-I_0)$ for a Ta 2nm/MnAu₂ 100nm device. The measurements were performed after the field-sweep chirality control. The magnetic field H_0 and electric current I_0 are used for the chirality control. $H_0 > 0$ in **a, c, e, g** and $H_0 < 0$ in **b, d, f, h**.

Comment(6): Please, add more details about the measurement procedure – for example, line 224 – the electrical current was DC? What was the amplitude (also 2e9 A/m²)? And how does the R_{1w} look? Why was not measured simultaneously panel Fig.5a and Fig.5c?

Response(6)

We have tried to describe the detail of the experiments as much as possible. The sentences around the 224th line in the previous version (224th also line in the present version) describe how the current-induced spin polarization can be observed as the transverse resistance. They explain the experimental concept rather than the detailed condition. In principle, the transverse resistance can be measured with DC electrical current, but we measured it with AC electric current for sensitive measurement as already described in the Methods section. We measured 1-omega AC voltage and deduced the 1-omega transverse resistance, dividing it by the AC current. The amplitude of AC current was 2 mA. “ $R_{1\omega}$ ”, which seems the longitudinal resistance, is shown above. We simultaneously measured the data shown in Fig. 5a and Fig. 5c.

Comment (7) line 245: why would Hall and planar Hall arise from misalignment of the voltage electrodes?

Response(7): Our statement is “The main contribution to R_T seems to be from trivial effects such as longitudinal resistance, the Hall effect, and planar Hall effect arising from the misalignments of voltage electrodes and the magnetic field.” Therefore, the longitudinal resistance and misalignment of magnetic fields are also related to this statement. The Hall signal is caused by the misalignment of a magnetic field. If there is a finite out-of-plane component of the magnetic field, the conventional Hall effect gives rise to transverse resistance. The planar Hall effect can be caused by the angular misalignment of electrodes; the anisotropy induces the transverse voltage when the direction of electrodes deviates from the prescribed crystal axis.

Comment(8): wording could be improved, example: line 223-224 ‘Pt layer deposited on MnAu2 to a thickness 10nm’

Response(8): We have tried to correct wording errors in this paper including the one reviewer suggested.

Comment(9): Fig.5h – not sure if one can call it saturating – maybe measure more points above 8mA or between 6 and 8mA)?

Response(9): The main point in this figure is not the saturation but that ΔR_T increases in parallel with $\Delta R^{2\omega}_{asym}$ shown in Fig. 5g. To address this point, we have modified the sentence at the 262nd-263rd line. The parallel relationship is nicely presented in Fig. 5i.

Therefore, we believe that the additional experiment is not needed.

Comment(10): SI Fig. 10 – ΔR_T and $\Delta R^{2\omega}_{asym}$ – why are they changing sign at 300K?

Response(10): As already suggested in the figure caption, the origin remains to be clarified. The similarity of ΔR_T and $\Delta R^{2\omega}_{asym}$ implies that the mechanisms of these two phenomena are microscopically related. The information should be useful for future theoretical works elucidating their mechanisms.

Comment(11): SI Fig. 6 – the y scale should be zoomed around the data to clearly show the variation

Response (11): The purpose of this figure is to show negligible sample damage during the switching process. For this purpose, this presentation is appropriate. If the data is magnified, variation owing to the temperature fluctuation is pronounced as shown below (Fig. R4).

Figure R4 Magnified view of Supplementary Fig. 6.

Comment(12): Is it necessary to advertise the results heavily in context of application (line 298)? It is very interesting fundamental research that has a long way to application and “chirality based magnetic memory” and many issues to solve (small signals,

competition temperature vs heat assisted poling procedure etc).

Response(12): We agree that this result is a very interesting fundamental research, and believe that it is good to show how such a fundamental research is potentially connected to future applications even if the distance to the application is not short.

--- Summary of the revisions made in the revised manuscript ---

Please stop the display of track change when you refer to the line number.

1. To respond to the reviewer #1's comment(1), we have added the magnitude of ac current for the 2nd harmonic resistivity measurements to the Methods section, and a short section regarding NET to the supplementary information(the 2nd section).
2. To respond to the reviewer #2's comment(1), We added a section regarding the theoretical calculation for the magnetic anisotropy dependence of chirality switching to the supplementary information (the 3rd section).
3. To respond to the reviewer #2's comment(1), we have added the description at the 279th-281st line in the main text.
4. To respond to the reviewer#2's comment(4), we have added the description at the 253rd-256th line.
5. To respond to the reviewer#2's comment(6), we have modified the Methods section.
6. To respond to the reviewr#2's comment(8), We have tried to correct wording errors in this paper including the one reviewer suggested.
7. To respond to the reviewr#2's comment(9), we have modified the sentence at the 262nd-263rd line.

Reviewers' Comments:

Reviewer #1:

Remarks to the Author:

In my opinion, the responses provided by the authors are convincing, and the manuscript is appropriate for publication in Nature Communications.

Reviewer #2:

Remarks to the Author:

I have read the revised version of the manuscript of Masuda et al. and the response letter. Overall minimal changes were made in the revised version of the manuscript and the requested "sanity checks" measurements were not done. The Response Letter alone did not convince me about both the novelty and the unambiguity of the MnAu₂/Pt experiments (details are following). I, therefore, still cannot recommend the manuscript for publication in Nature Communication and I would recommend a more specialized journal.

Novelty: The authors responded that they see novelty in 3 points: 1) high transition temperature, 2) chirality switching and 3) zero field chirality switching.

1) Higher transition temperature is only a material choice and in my opinion does not justify publication in a high impact journal. It might be a milestone in other research directions, however, in the case of this manuscript I see a different roadblock for further increase: the chirality switching requires accessible critical temperature, therefore, the application potential seems to be limited.

The statement from the Response Letter: "there is some heating effect but traversing the transition temperature is not mandatory for the process of chirality switching" is not convincing. First, the exact local temperature during highly non-equilibrium pulsing experiments might be difficult to estimate. And most importantly, both the need of crossing the critical temperature or heating the sample close to the critical temperature represents the same problem for applications. The real breakthrough would be to show the switching well below the critical temperature.

In this context also the Discussion section is misleading – the problem for applications is not to find a material with higher critical temperature (by strain, doping etc), but to achieve switching without the need to heat it close to the critical temperature.

2) Chirality switching is achieved by combined effect of magnetic field and electrical current in the vicinity of the critical temperature. The physics of switching is in principle same if the pulsing or static current is used or magnetic field is ramped or static. I agree that it is nice to show that the switching is possible also by pulses of j , but it appears as a technical progress and does not justify publication in Nat.Comm. The same is true for the newly added calculation which suggests that weaker magnetic anisotropy reduced (expectedly) the critical switching current.

3) Zero field chirality detection – here I see the largest potential for novelty, however, more detailed and robust analysis should be done as I mentioned in the first report (and more details follow).

MnAu₂/Pt bilyaer experiments: Although the authors improved the description, how exactly the data are collected, they did not provide any new data. The authors repeatedly state that they are "quite confident", "quite sure" or "deduce the relation between the chirality and transverse resistance in a watertight way" I, however, believe that to report a new phenomenon in a high profile journal the authors should do maximum to exclude artifacts and understand the details. Unlike the authors I think that the reference measurement I am asking are related to the reliability of the data analysis. For example:

- the authors claim that a potential FM polarization cannot produce ΔR_T . However, in

principle the opposite current I_0 produces opposite Oersted field during poling and it could influence the interface (FM polarization or any other complex spin state of the interface). The resulting R_T would be than different for example via spin Hall magnetoresistance or magnetoresistance of the polarized layer alone and therefore also make a difference in the ΔR_T . Inserting Cu layer would exclude the polarization of Pt and also change the potential spin Hall magnetoresistance. In the same time a thin Cu layer should not affect the spin injection mechanism proposed by the authors.

- the authors state they have Ta capped sample – they could directly compare bilayers MnAu2/Pt and MnAu2/Ta given the opposite spin Hall angle of Ta and Pt they should see reversal of the ΔR_T

- the authors write that the small magnetic hysteresis is “not what they want to prove and it is just an obstacle”, but where the hysteresis comes from? Is this present only to the MnAu2/Pt bilayer? I do not understand the statement “transverse resistance is generally sensitive to magnetic anisotropy as well as uniform magnetization” – the SQUID magnetometry is not? Or do they mean the hysteresis arises from an interfacial effect or effect arising from shape anisotropy which is too small to be visible in SQUID? But small interfacial effect can influence the resulting signal – see above

--- Authors' response to Reviewer 2 ---

We would thank Reviewer 2 for the comments. What follows are our responses to each comment raised by Reviewer 2 and the corresponding revisions in our manuscript.

Reviewer's Comment (1): *I have read the revised version of the manuscript of Masuda et al. and the response letter. Overall minimal changes were made in the revised version of the manuscript and the requested "sanity checks" measurements were not done. The Response Letter alone did not convince me about both the novelty and the unambiguity of the MnAu₂/Pt experiments (details are following). I, therefore, still cannot recommend the manuscript for publication in Nature Communication and I would recommend a more specialized journal.*

Response(1): At this time, we certainly did the "sanity checks" measurements with hard work in three months. As detailed below, the experimental results do support our scenario. We believe that the reviewer will agree with the publication in Nature Communications.

Reviewer's Comment(2):

1) Higher transition temperature is only a material choice and in my opinion does not justify

publication in a high impact journal. It might be a milestone in other research directions, however, in the case of this manuscript I see a different roadblock for further increase: the chirality switching requires accessible critical temperature, therefore, the application potential seems to be limited.

The statement from the Response Letter: "there is some heating effect but traversing the transition temperature is not mandatory for the process of chirality switching" is not convincing. First, the exact local temperature during highly non-equilibrium pulsing experiments might be difficult to estimate. And most importantly, both the need of crossing the critical temperature or heating the sample close to the critical temperature represents the same problem for applications. The real breakthrough would be to show the switching well below the critical temperature.

In this context also the Discussion section is misleading – the problem for applications is not to find a material with higher critical temperature (by strain, doping etc), but to achieve switching without the need to heat it close to the critical temperature.

2) Chirality switching is achieved by combined effect of magnetic field and electrical current in the vicinity of the critical temperature. The physics of switching is in principle same if the pulsing or static current is used or magnetic field is ramped or static. I agree that it is nice to show that the switching is possible also by pulses of j , but it appears as a technical progress

and does not justify publication in Nat.Comm. The same is true for the newly added calculation which suggests that weaker magnetic anisotropy reduced (expectedly) the critical switching current.

Response(2): We still believe that the room temperature chirality control and the achievement of chirality switching are novel enough and deserve publication in Nature Communications. Nevertheless, further arguments at these points may not be constructive because we already explained our opinion a lot in the previous round. Instead, we would like to emphasize that the fabrication of helimagnetic epitaxial thin film with a controllable chirality should pave an avenue for helimagnetic spintronics with complex thin film devices. We believe that the reviewer will also appreciate the importance of thin-film fabrication after admitting the validity of the chirality-dependent transverse resistance experiments on the multilayer films.

In addition, we would like to suggest a small point related to the reviewer's suggestion "*the exact local temperature during highly non-equilibrium pulsing experiments might be difficult to estimate.*" In principle, temperature cannot be defined in such a non-equilibrium state. The effective average energy of lattices should be quite different from that of electrons in the large current density. What we show in the switching experiment is that the non-equilibrium

state is useful for the chirality control.

Reviewer's comment (3)

Zero field chirality detection – here I see the largest potential for novelty, however, more detailed and robust analysis should be done as I mentioned in the first report (and more details follow).

MnAu2/Pt bilayer experiments: Although the authors improved the description, how exactly the data are collected, they did not provide any new data. The authors repeatedly state that they are “quite confident”, “quite sure” or “deduce the relation between the chirality and transverse resistance in a watertight way” I, however, believe that to report a new phenomenon in a high profile journal the authors should do maximum to exclude artifacts and understand the details. Unlike the authors I think that the reference measurement I am asking are related to the reliability of the data analysis. For example:

- the authors claim that a potential FM polarization cannot produce ΔR_T . However, in principle the opposite current I_0 produces opposite Oersted field during poling and it could influence the interface (FM polarization or any other complex spin state of the interface). The resulting R_T would be than different for example via spin Hall magnetoresistance or

magnetoresistance of the polarized layer alone and therefore also make a difference in the ΔR_T . Inserting Cu layer would exclude the polarization of Pt and also change the potential spin Hall magnetoresistance. In the same time a thin Cu layer should not affect the spin injection mechanism proposed by the authors.

Response(3):

We certainly did the chirality detection measurement for the Pt/Cu/MnAu₂ trilayer thin film sample and observed the even enhanced chirality-dependent transverse resistance as shown in Supplementary Fig. 13. The reviewer seemed to worry that the ferromagnetic polarization at the interface with Pt might induce ΔR_T but this experiment clearly shows this is not the case.

Reviewer's comment (4)

- the authors state they have Ta capped sample – they could directly compare bilayers MnAu₂/Pt and MnAu₂/Ta given the opposite spin Hall angle of Ta and Pt they should see reversal of the ΔR_T

Response(4)

Because the quality of the Ta layer was not good judging from the X-ray diffraction profile, we chose W as the top layer material instead of Ta. The sign of spin Hall angle of W is opposite to that of Pt, and the same as that of Ta. We confirmed that the chirality-dependent transverse resistance of the W/MnAu₂ bilayer sample is opposite to that of the Pt/MnAu₂ bilayer sample as shown in Supplementary Fig. 14.

Reviewer's comment (5)

- the authors write that the small magnetic hysteresis is "not what they want to prove and it is just an obstacle", but where the hysteresis comes from? Is this present only to the MnAu₂/Pt bilayer? I do not understand the statement "transverse resistance is generally sensitive to magnetic anisotropy as well as uniform magnetization" – the SQUID magnetometry is not? Or do they mean the hysteresis arises from an interfacial effect or effect arising from shape anisotropy which is too small to be visible in SQUID? But small interfacial effect can influence the resulting signal –

The transverse resistance is induced by the off-diagonal component of the resistivity tensor. What the quantity probes is quite different from what uniform magnetization does. For example, the transverse resistance would sensitively probe the tilting of a helical plane in the

helimagnetic state because it should affect the anisotropy of resistivity, but it does not affect the uniform magnetization. Therefore, we think that it is not unnatural that the hysteresis appears only in the transverse resistance while we did not identify the definite origin of the hysteresis.

--- Summary of the revisions made in the revised manuscript ---

1. We have added the Supplementary Figs. 13 and 14 to the Supplementary Information, which show the results of the additional experiments requested by the reviewer 2.
2. We have added the description at the 266th-272nd lines in the main text, explaining the results of the additional experiments.

Reviewers' Comments:

Reviewer #2:

Remarks to the Author:

I have reviewed the revised manuscript by Masuda et al. In this second round of revisions, the authors have made significant efforts, including conducting control experiments. These additions have strengthened the results pertaining to zero field switching, making the manuscript more convincing and significantly improved. From a technical standpoint, I think the manuscript is suitable for publication, with the aspect of zero field switching being particularly intriguing. However, I maintain a slight skepticism regarding its novelty for the reasons outlined below. This perspective might be subjective, and if the other referees and the editor perceive the work as sufficiently novel, then my concerns should be outweighed.

Concerns Regarding Novelty:

Chirality Switching: This phenomenon has been previously reported.

Room Temperature Significance: I am not convinced regarding the importance of this aspect as I detailed in previous reports.

Thin Film Fabrication: The growth of single-phase MnAu₂ films appear to have been reported earlier (<https://www.sciencedirect.com/science/article/pii/S0304885316316481>)

Comparison of Single Crystals to Thin Films: While important for device fabrication, this does not seem sufficient to justify publication in a journal like Nature Communications.

Switching by Pulses or DC Current: This seems more of a technical advancement rather than a novel scientific discovery.

We would thank Reviewer 2 for reviewing our revised manuscript. What follows are our responses to comments raised by Reviewer 2.

Reviewer's Comment: *I have reviewed the revised manuscript by Masuda et al. In this second round of revisions, the authors have made significant efforts, including conducting control experiments. These additions have strengthened the results pertaining to zero field switching, making the manuscript more convincing and significantly improved. From a technical standpoint, I think the manuscript is suitable for publication, with the aspect of zero field switching being particularly intriguing.*

However, I maintain a slight skepticism regarding its novelty for the reasons outlined below. This perspective might be subjective, and if the other referees and the editor perceive the work as sufficiently novel, then my concerns should be outweighed.

Concerns Regarding Novelty:

Chirality Switching: This phenomenon has been previously reported.

Room Temperature Significance: I am not convinced regarding the importance of this aspect as I detailed in previous reports.

Thin Film Fabrication: The growth of single-phase MnAu₂ films appear to have been reported earlier (<https://www.sciencedirect.com/science/article/pii/S0304885316316481>)

Comparison of Single Crystals to Thin Films: While important for device fabrication, this does not seem sufficient to justify publication in a journal like Nature Communications.

Switching by Pulses or DC Current: This seems more of a technical advancement rather than a novel scientific discovery.

Response:

First, we would appreciate the reviewer's fair decision and thoughtful comments to admit the publication although they have slight skepticism about the novelty.

As the reviewer listed, this paper represents many accomplishments. The reviewer decomposed them and was concerned about the novelty of each part. However, all things considered, our paper has quite enough novelty. With these accomplishments, our paper clearly demonstrates the spintronics based on helimagnets utilizing the chirality degree of freedom is feasible at room temperature for the first time. Therefore, we strongly believe that our paper deserves publication in Nature Communications.

In addition, we would like to suggest a small point regarding the comment on the thin film

fabrication.

Thin Film Fabrication: The growth of single-phase MnAu₂ films appear to have been reported earlier (<https://www.sciencedirect.com/science/article/pii/S0304885316316481>)

The reported single-phase MnAu₂ films are polycrystalline or have multiple domains. The epitaxial, single crystalline film was fabricated for the first time in our present research. This was achieved by choosing the proper substrate, that is, hexagonal ScMgAlO₄ (10–10) substrate with the in-plane anisotropic crystal structure. Since the epitaxial film is indispensable for the chirality control experiments in the present magnetization configuration, we believe that this is a sufficient advance in helimagnet-based spintronics. To clarify this point, we added the description at 326th - 328th lines in the Methods section and newly added ref. 53.